# Construction of Power Fault Knowledge Graph Based on Deep Learning

**Peishun Liu** [1] , **Bing Tian** [2] , **Xiaobao Liu** [2] , **Shijing Gu** [1,*], **Li Yan** [2] , **Leon Bullock** [1,*] , **Chao Ma** [2] , **Yin Liu** [2] **and Wenbin Zhang** [2]

1   College of Information Science and Technology, Ocean University of China, Qingdao 266100, China; liups@ouc.edu.cn
2   Information & Telecommunications Company, State Grid Shandong Electric Power Company, Jinan 250001, China; tianbing@sd.sgcc.com.cn (B.T.); liuxiaobao@sd.sgcc.com.cn (X.L.); yanli@sd.sgcc.com.cn (L.Y.); machao@sd.sgcc.com.cn (C.M.); liuyin@sd.sgcc.com.cn (Y.L.); wbzhang@sd.sgcc.com.cn (W.Z.)
*   Correspondence: gushijing@stu.ouc.edu.cn (S.G.); leonbevanbullock@ouc.edu.cn (L.B.)

**Abstract:** A knowledge graph can structure heterogeneous knowledge in the field of power faults, construct the correlation between different pieces of knowledge, and solve the diversification, complexity, and island of fault data. There are many kinds of entities in power fault defect text, the relationship between entities is complex, and the data are often mixed with noise. It is necessary to research how to effectively mine the target data and separate the salient knowledge from the noise. Moreover, the traditional entity and relationship extraction methods used in the construction of a power fault knowledge graph cannot fully understand the text semantics, and the response accuracy is low. The Log system usually contains all kinds of information related to faults and a log analysis helps us collect fault information and perform association analysis. Therefore, a Bidirectional Sliced GRU with Gated Attention mechanism (BiSGRU-GA) model is proposed to detect the anomalous logs in the power system, this enriches the fault knowledge base and provides a good data resource for the construction of the knowledge graph. A new Bidirectional GRU with Gated Attention mechanism and Conditional Random Fields and a BERT input layer (BBiGRU-GA-CRF) model is proposed by introducing a BERT layer and Attention Mechanism into the Bidirectional GRU (BiGRU) model to more fully understand the context information of fault sentences and improve the accuracy of entity recognition of fault sentences. Aiming to solve the problems of large calculation cost and propagation error which occur in the traditional relationship extraction model, an improved Bidirectional Gated Recurrent Unit neural network with fewer parameters and the Gated Attention Mechanism (BiGRU-GA) model is proposed. This new model introduces an improved Gated Attention Mechanism to achieve better effects in relationship extraction. Compared with Bidirectional Long Short-Term Memory with Attention Mechanism (BiLSTM-Attention), the accuracy, recall, and F-measure of the model were improved by 1.79%, 13.83%, and 0.30% respectively, and the time cost is reduced by about 16%. The experimental results show that the BiGRU-GA model can capture local features, reduce the training time cost, and improve the model recognition effect.

**Keywords:** power failure; knowledge graph; attention mechanism; GRU; BERT

## 1. Introduction

### 1.1. Research Progress of Power System Fault Processing Technology

With the increase in the complexity and type of the power system, the number of faults in China's power system is increasing. It poses a huge threat to the production, operation, maintenance, and security of China's power system and the stable development of the national economy. At present, the power system is moving in the direction of intelligence and automation. A large number of sensor devices are used for equipment condition

monitoring. It is of great significance to ensure the safe and stable operation of the power grid to mine the information related to equipment status from the accumulated massive data and judge whether there is a fault or predict a fault. The related technologies of power system fault diagnosis have been developed rapidly. Various methods are used to diagnose power system faults from different perspectives. The fault diagnosis method can be divided into expert system [1,2], artificial neural network [3,4], Bayesian network [5], Petri net technology [6], analytic model [7,8], information fusion [9,10], and other methods.

The expert system simulates the knowledge and experience of experts to deduce and judge, and finally obtain the explanation closest to the actual situation. With the continuous expansion of the power system scale, the construction and maintenance of knowledge base has become increasingly complex. When information is missing or confused, the accuracy of expert system diagnosis is greatly reduced [1,2].

Artificial Neural Network (ANN) is a technology that can process information by simulating the human nervous system. The ANN obtains fault diagnosis results through information input, but it is difficult to complete the explanation of the whole fault development process [3]. In recent years, the ANN have been widely used in fault area division and fault location, which greatly improved learning efficiency. However, its interpretation ability in the fault diagnosis of a large power grid needs to be further improved [4].

The Bayesian network can reasonably explain the behavior of the system and can express and reason knowledge in the presence of uncertain factors. Under the condition that complete information can be provided to a Bayesian network it provides accurate fault diagnosis, but in a single information event, or in the case where there are omissions or errors in the information, this method cannot accurately diagnose a fault. This method needs to quantify the probabilities of collected information, process the uncertain information, then through using pattern recognition, establish a correlation Bayesian network model to determine fault components and fault locations [5].

The fault diagnosis method of Analytic Models is mainly based on the switching information of power systems. The action and alarm information of the protection device are analyzed, and the optimal solution is solved by intelligent algorithm [7]. In the process of solving the model, there are multiple optimal solutions, and the most accurate diagnosis is difficult to obtain [8].

Information fusion is a new technology for fault diagnosis in recent years, and it is also an important development direction for the future. The main advantage of this method is that it can analyze power system faults from multiple angles and avoid diagnosis errors caused by a single point of information or information loss [9]. With the continuous improvement of big data technology, power system fault diagnosis based on multi-source information fusion has also achieved certain development [10].

In the case of complex faults of a power system, there are many uncertain factors, which lead to some errors in the above diagnosis methods. At present, the main countermeasures for the complex fault diagnosis of power systems still rely on the experience of operation and maintenance personnel. Manually determining the fault type and providing the fault solution greatly depends on previous knowledge. This restricts the efficiency and accuracy of complex fault handling.

*1.2. Power System Fault Processing Technology Based on Deep Learning Technology*

In recent years, with the rapid development of computer technology, the research tide of deep learning technology has been rising. Deep learning technology can process large amounts of data and extract deep features of input data. It has been applied to healthcare, image recognition and natural language processing with great success [11]. The application of deep learning technology in fault diagnosis can realize the combination of feature extraction and classification, and has less dependence on professional knowledge. Deep learning is widely used in power systems, such as power consumption prediction, wind and solar energy prediction, power disturbance detection and classification, fault detection, energy management and energy optimization [12–14]. Deep learning technology

has strong abilities of data feature extraction and provides a new research direction for power system fault diagnosis [15].

In 2012, Google took the lead in introducing the concept of knowledge graphs. As a special knowledge base, a knowledge graph extracts entities and relationships from unstructured knowledge to form relationships, and stores these relationships in the form of a directed graph, which can express unstructured knowledge in a standardized way. A knowledge graph can be regarded as a semantic map composed of several nodes and edges. Each entity or concept is a node in the knowledge graph [16]. The concept is the abstraction of a certain kind of phenomenon or transaction, such as the system Operation and Maintenance personnel; an entity is a specific person or thing, such as the specific name of the system operation and maintenance personnel. The edge connecting two nodes represents a certain relationship between the nodes. Among many different knowledge representation methods, the two characteristics of knowledge graphs are particularly prominent, that is, strong expressive ability and ease of extension. Utilizing the powerful expressive ability of a knowledge graph and its rich relationship information, then combining it with the relationship reasoning rules in the scenario of power system production, operation and maintenance, the log data can be mined to complete various reasoning applications, such as inconsistency detection, inference completion, knowledge discovery, auxiliary reasoning, and decision-making.

Knowledge graphs can structure the heterogeneous knowledge and construct the correlation between pieces of knowledge. They can be used to solve the diversification, complexity and islanding of fault data, and standardize the storage of fault knowledge for fault diagnosis. In addition, the knowledge graph also has the advantage of interpretability, which can provide a reasonable explanation for the fault causes and solutions. At present, knowledge graph technology is not widely used in the field of fault diagnosis. Li Jinxing et al. [17] studied fault treatment based on power information knowledge graphs and proposed a fault diagnosis method. A BiLSTM-CRF model was used to conduct entity recognition of power domain knowledge and realize the construction and application of a power grid domain knowledge graph. Feng et al. [18] proposed an intelligent question answering system for fault diagnosis of power information acquisition systems based on knowledge graph technology, which can efficiently traverse searching of nodes and paths thus significantly improve reasoning efficiency. Meng et al. [19] proposed the BERT–BiLSTM–CRF model to extract knowledge from Chinese technical literature and construct a knowledge graph of electric power equipment faults.

In power equipment fault defect text, there are many entities related to equipment fault information, with various categories and large differences in features. It is difficult to extract all entities in the text by using a single dictionary or machine learning method, some entity values are expandable, so there is no directly applied extraction method. In addition, there is not much research on text processing in this field, and there is still a lack of universal text data sets, which need to be made by ourselves. Due to the diversity of entities in power equipment fault defect text, the relationships between entities will be more complicated, which greatly increases the difficulty of accurately judging the relationship between entities. A log system usually contains a large variety of information related to faults and a log analysis helps researchers collect fault information and perform association analysis. Therefore, making full use of the fault prior knowledge in the power field, integrating the abnormal logs generated in the operation of the power system, and constructing a perfect and practical power system fault knowledge graph is of great significance to improving the power system fault handling capacity and level of intelligence.

At present, the construction of the power grid fault knowledge graph is mainly based on the distribution network equipment account data, fault handling data, dispatching regulation data, and distribution network defect data as objects, most of which are unstructured data written in Chinese. In the process of Chinese knowledge graph construction, the word segmentation system can directly realize the boundary division of most entities and concepts, so the performance of the word segmentation system plays a key role in

the construction of a knowledge graph. The performance of Chinese word segmentation systems in standard data sets (such as news corpus) is nearly perfect, but the performance of the Chinese word segmentation system in other specific fields without annotations is not satisfactory, especially in some professional fields, because there are a large number of unknown words. In order to solve this problem, this paper introduces logs in the system as the source of fault data. Logs are semi-structured data with a large quantity of rich information, which is convenient for automatic processing. A new BBIGRU-GA-CRF model is proposed. The model firstly identifies and extracts the power equipment entities from the pre-processed logs and Chinese technical literature, and then extracts the semantic relationships between the entities using the relationship classification method based on dependency parsing. Finally, the extracted knowledge is stored in the form of triples in the Neo4j database and visualized in the form of graphs. Through the above steps, the Chinese knowledge graph of power equipment fault is established.

### 1.3. Major Contribution and Organizatio

The main contributions of this paper are as follows:

1. A power system log anomaly detection model based on BiSSGRU-GA is proposed, which accurately detects the abnormal logs generated during the operation of the power system and analyzes the abnormal or fault information as a fault data source. This enriches the data sources needed to build knowledge graphs;

2. An entity extraction model based on BBiGRU-GA-CRF is proposed, which can fully extract the characteristics at the word level and dynamically adjust the vector representation of words and sentences according to the contextual semantic environment, so that it can accurately express the meaning in the current environment, improve the generalization ability of the model, and enhance the effect of entity recognition;

3. A relationship extraction model based on BiGRU-GA is proposed, which introduces an improved Gated Attention Mechanism, efficiently utilizes vector features, captures local features, reduces training time overhead, and enhances the effect of entity recognition.

Section 2 provides more details on the methods for GRU and GA.

Section 3 introduces the three different models: BiSGRU-GA is a log anomaly detection model used to obtain fault logs from power system logs, BBiGRU-GA-CRF is an entity extraction model, and BiGRU-GA is a relation extraction model.

Section 4 shows experiments conducted with each method, where they are each compared with several peers.

Section 5 discusses how the work can be further extended.

## 2. Overview of GRU and GA

This section describes the theory of GRU and GA used in this paper.

### 2.1. Introduction to Gated Recurrent Units (GRU)

GRU inherits the advantages of Long Short-Term Memory (LSTM) networks and simplifies its structure while retaining high efficiency. Compared with LSTM, GRU only contains two gates: the update gate and reset gate. Therefore, GRU has recently become a widely used neural network. The GRU structure is shown in Figure 1. GRU and LSTM have the same function when capturing long sequence semantic association, GRU can effectively suppress gradient disappearance or explosion, and the effect is superior to traditional RNNs and the computational complexity is lower than LSTM. The deep neural network model in this paper is mainly based on GRU.

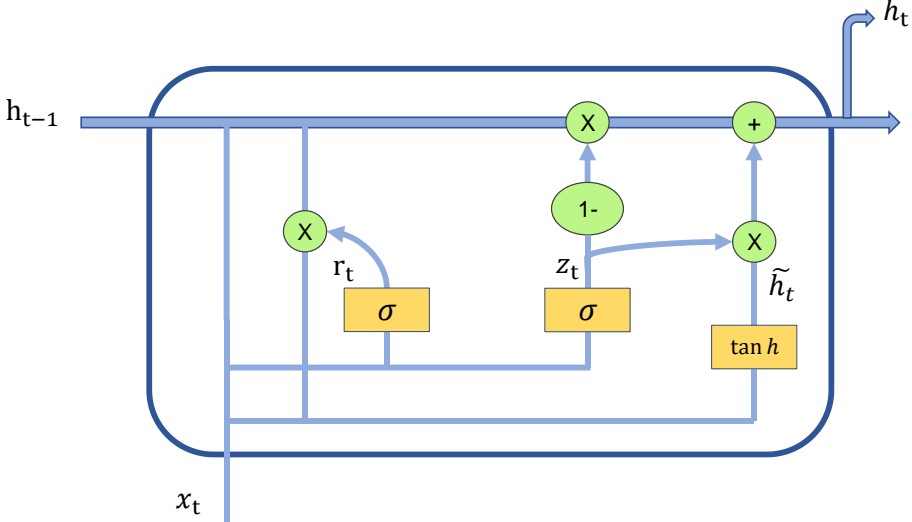

**Figure 1.** Standard GRU structure.

In the GRU structure, the update gate is set as $z_t$ and the reset gate is set as $r_t$. The formula of GRU networks is shown in the Equations (1)–(5):

$$r_t = \sigma(w_r \cdot [h_{t-1}, x_t]) \tag{1}$$

$$z_t = \sigma(w_z \cdot [h_{t-1}, x_t]) \tag{2}$$

$$\widetilde{h_t} = tanh(w_{\widetilde{h}} \cdot [r_t * h_{t-1}, x_t]) \tag{3}$$

$$h_t = (1 - z_t) * h_{t-1} + z_t * \widetilde{h_t} \tag{4}$$

$$y_t = \sigma(w_0 \cdot h_t) \tag{5}$$

### 2.2. Introduction to Gated Attention Mechanism (GA)

The Attention Mechanism is used to provide corresponding weight values to different log sequences according to the importance of the sequence. However, the traditional Attention Mechanism network assigns weight to each input unit, and even irrelevant units have some weight, which leads to the attention weight of relevant units becoming particularly small; for long sequences, the performance degradation caused by it cannot be ignored. Lanqing Xue et al. [20] integrated the gating idea into the Attention Mechanism to calculate the selected elements, while the unselected elements did not participate in the calculation. This allows the model to pay differential attention to different parts of the sequence, which greatly improves the performance of the model by reducing noise. The attention network structure based on gating is shown in the figure below, including secondary network and backbone network. It is shown in Figures 2 and 3:

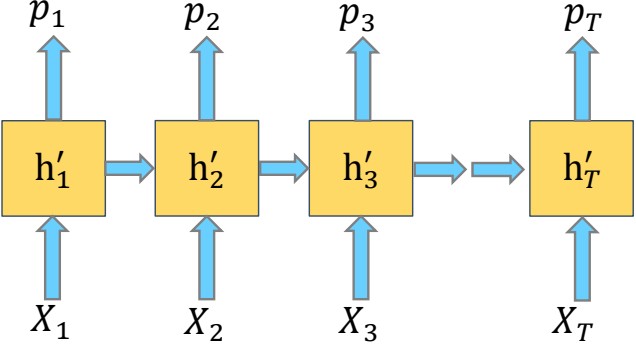

**Figure 2.** Auxiliary network.

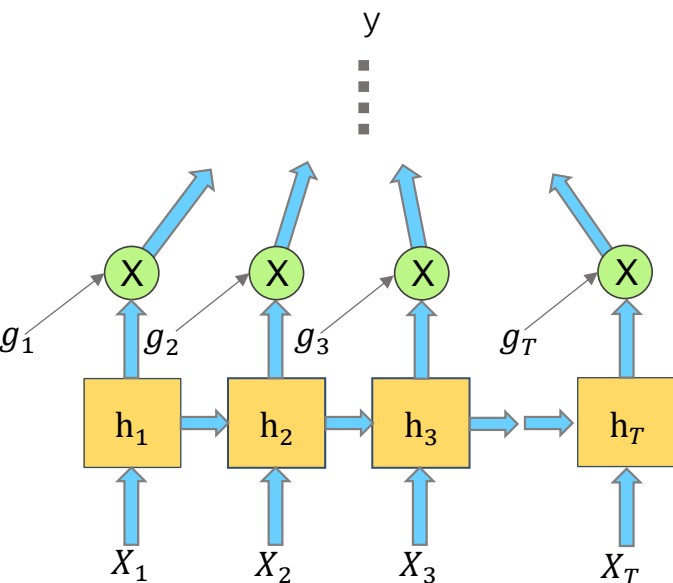

**Figure 3.** Backbone network.

GA uses an auxiliary network to dynamically select a subset of elements to participate in and compute attention weights to aggregate the selected elements, requiring less computation and better interpretation. The Attention Mechanism in this paper is constructed based on GA.

## 3. Construction Model of Power Fault Knowledge Graph Based on Deep Learning

Entity extraction is an important part of fault recovery knowledge graph construction. Existing entity extraction methods require a large amount of labeled corpus. If part of the labeled corpus is reduced, the extraction accuracy of the method is severely reduced. This section introduces the power system log anomaly detection model based on the bidirectional slice gated cyclic unit and Gated Attention Mechanism (BiSGRU-GA), power fault entity extraction model based on BBiGRU-GA-CRF, and power fault relation extraction model based on BiGRU-GA. Based on the above three models, a power fault knowledge graph system is constructed. The method presented in this paper reduces the model's dependence on labeled corpus, which is of significance for improving the power grid fault processing capability.

### 3.1. Bidirectional Sliced Gated Recurrent Units with a Gated Attention Mechanism (BiSGRU-GA) Log Anomaly Detection Model

Log anomaly detection can be attributed to sequence classification or prediction. Because it is very difficult to directly process unstructured log data, the usual method of log anomaly detection is to obtain log templates through log parsing, and then perform anomaly detection on the templates. The template is equivalent to the "summary" of the log, and the log can be obtained as a template plus parameters [21]. A log key sequence can be regarded as a template and based on the ideas introduced in DeepLog [22], a new method was designed and implemented; it is a neural network model named Bidirectional Sliced GRU with a Gated Attention Mechanism [23] (BiSGRU-GA). This model has fewer parameters and a faster convergence speed than LSTM models. Whilst the purpose was to reduce the training costs, an additional benefit was that the accuracy was also greatly improved. The model structure is shown in Figure 4. We take the input length of 8 and the minimum subsequence of 2 as an example:

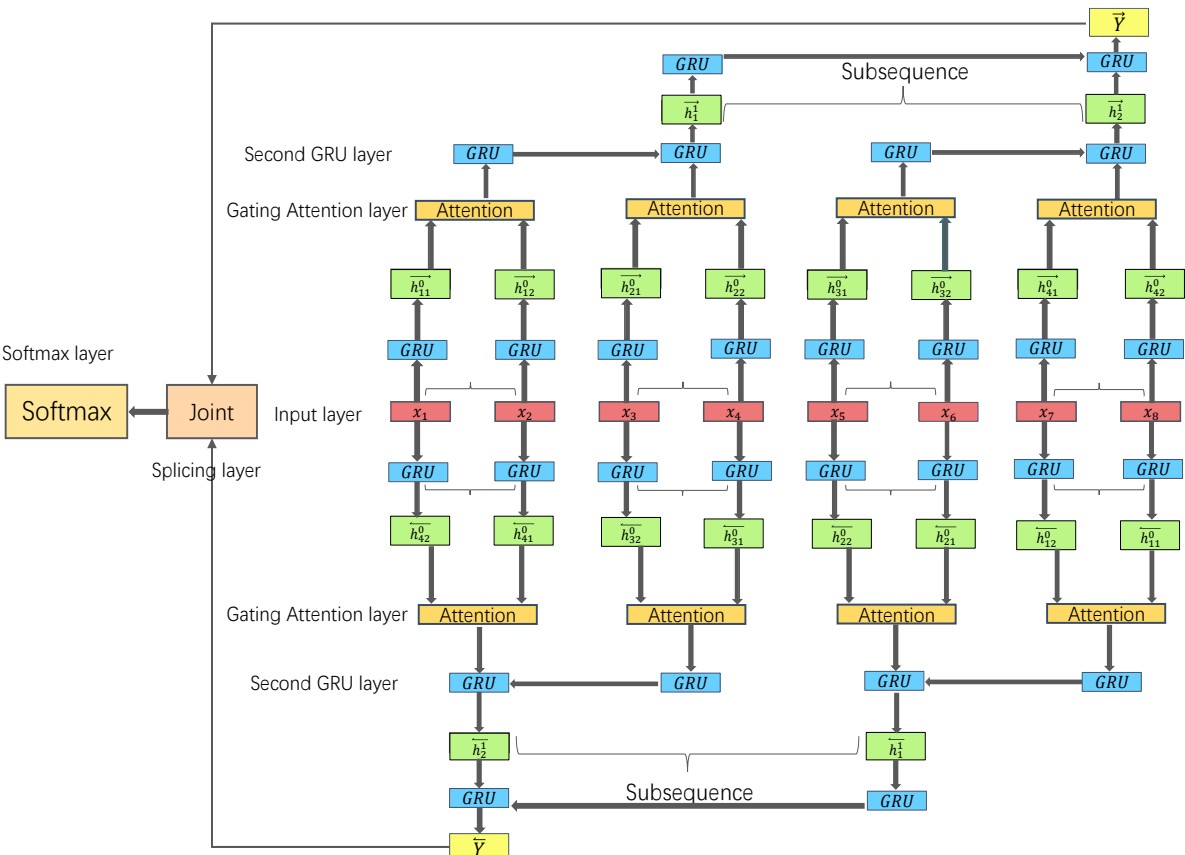

**Figure 4.** Log anomaly detection BiSGRU-GA.

First, the word2vec method is used to generate the log key word vector $x_m$ from the system log. Thus, the length of each log key is composed of max key word vectors and $x_{ij}$ is used to represent $X_i$. The expression of $X_i$ is as shown in Equation (6):

$$X_i = x_{i1} \oplus x_{i2} \oplus \ldots x_{ij} \tag{6}$$

Then the sequence generated by the word2vec is input into the GRU layer in both forwards and backwards directions. The specific calculation formula is as shown in the Equations (7) and (8):

$$\overrightarrow{h_{ij}^o} = \overrightarrow{GRU^0}(x_{ij}, \overrightarrow{h_{i(j-1)}^0}) \tag{7}$$

$$\overleftarrow{h_{lj}^o} = \overleftarrow{GRU^0}(x_{ij}, \overleftarrow{h_{l(J-1)}^0}) \tag{8}$$

where $x_{ij}$ represents the *j*-th input of the *i*-th smallest subsequence in the processing direction, $\overrightarrow{h_{ij}^o}$ is a vectorization status after GRU processing in the forwards direction and $\overleftarrow{h_{lj}^o}$ is a vectorization status after GRU processing in the backwards direction

The Attention layer is the core part of the network model BiSGRU-GA. This layer assigns weight to the filtered words in each log key. The representation of each minimal subsequence $s_i$ is represented by the weighted sum of vectors so as to realize the focus of some location key features at the log key level.

Then the output sequence of the attention layer in the second GRU layer is also processed in two directions. The specific calculation formula is as shown in the Equations (9) and (10):

$$\overrightarrow{h_t^1} = \overrightarrow{GRU^1}\,(S_{t*\frac{p_0}{p_1}-l_0+1} \sim S_{t*\frac{p_0}{p_1}})  \tag{9}$$

$$\overleftarrow{h_t^1} = \overleftarrow{GRU^1}\,(S_{t*\frac{p_0}{p_1}-l_0+1} \sim S_{t*\frac{p_0}{p_1}})  \tag{10}$$

Then in the same way as the second GRU layer, the representation of the nth layer is output. The specific calculation is as shown in the Equations (11) and (12):

$$\overrightarrow{h_t^n} = \overrightarrow{GRU^n}\,(\overrightarrow{h}_{t*\frac{p_{n-1}}{p_n}-l_n+1} \sim \overrightarrow{h}_{t*\frac{p_{n-1}}{p_n}})  \tag{11}$$

$$\overleftarrow{h_t^n} = \overleftarrow{GRU^n}\,(\overleftarrow{h}_{t*\frac{p_{n-1}}{p_n}-l_n+1} \sim \overleftarrow{h}_{t*\frac{p_{n-1}}{p_n}})  \tag{12}$$

where $h_t^n$ is the implicit representation of the $t$-th subsequence of the nth layer GRU; $p_n$ is the number of N-level subsequences; and $l_n$ is the length of the $n$th subsequence.

Then the top layer $\overrightarrow{Y} = \overrightarrow{h_1^k}$ and $\overleftarrow{Y} = \overleftarrow{h_1^k}$ to splice $\overrightarrow{Y}$ and $\overleftarrow{Y}$ is calculated in the splicing layer and the result is input to the softmax layer. The mathematical form of the calculation process is as shown in Equation (13):

$$Y = \text{joint}\,(\overrightarrow{Y},\ \overleftarrow{Y})  \tag{13}$$

From this, the probability of the occurrence of each log key can be obtained, as shown in Equation (14):

$$p(x_t\,|C) = \frac{e^{(h_t V'(x_t))}}{\sum_{x_i \in V} e^{(h_t V'(x_i))}}  \tag{14}$$

where C represents the previously entered sequence, $x_t$ represents the log key, and $V'(x)$ is the output word vector in the softmax layer [23].

The loss function is as shown in Equation (15):

$$Loss = -\sum log\,p_{dj}  \tag{15}$$

Each log key carries a probability value. The log keys are arranged in descending order according to the probability value, and the log keys with the highest probability ranking are selected to form a trusted set. If the log key of the output log is in the trusted set, the log is regarded as a normal log, otherwise the log is regarded as an exception log.

The BiSGRU-GA log anomaly model is proposed to improve the training speed and strengthen the local attention.

### 3.2. Entity Extraction Model Based on BBiGRU-GA-CRF

Both the BiLSTM entity extraction model and the BiLSTM-CRF model can capture pre and post two-way information, but its model structure is slightly complex and requires more parameters during training, resulting in a long training time. The BiGRU-CRF model simplifies the structure of the BiLSTM model and BiLSTM-CRF model, requires fewer parameters for training, and achieves faster training speed. However, the BiGRU-CRF model cannot extract global features, which makes it difficult for the model to screen important information, resulting in low accuracy of model recognition and poor overall recognition effect.

To solve the above problems, this paper proposes an improved entity extraction model composed of four parts, BERT Bidirectional GRU with Gated Attention Mechanism, and

Conditional Random Fields (BBiGRU-GA-CRF). With the powerful text feature representation ability of the BERT model, the power fault text is represented, the global features are extracted, and the corresponding semantic vector sequences are obtained as the input of the BiGRU network layer. At the same time, the Gated Attention Mechanism is introduced to further mine the local features of fault text. Finally, the CRF module is used to complete the sequence decoding and annotation, and the optimal annotation sequence is obtained.

The BBiGRU-GA-CRF entity recognition model proposed in this paper is shown in Figure 5.

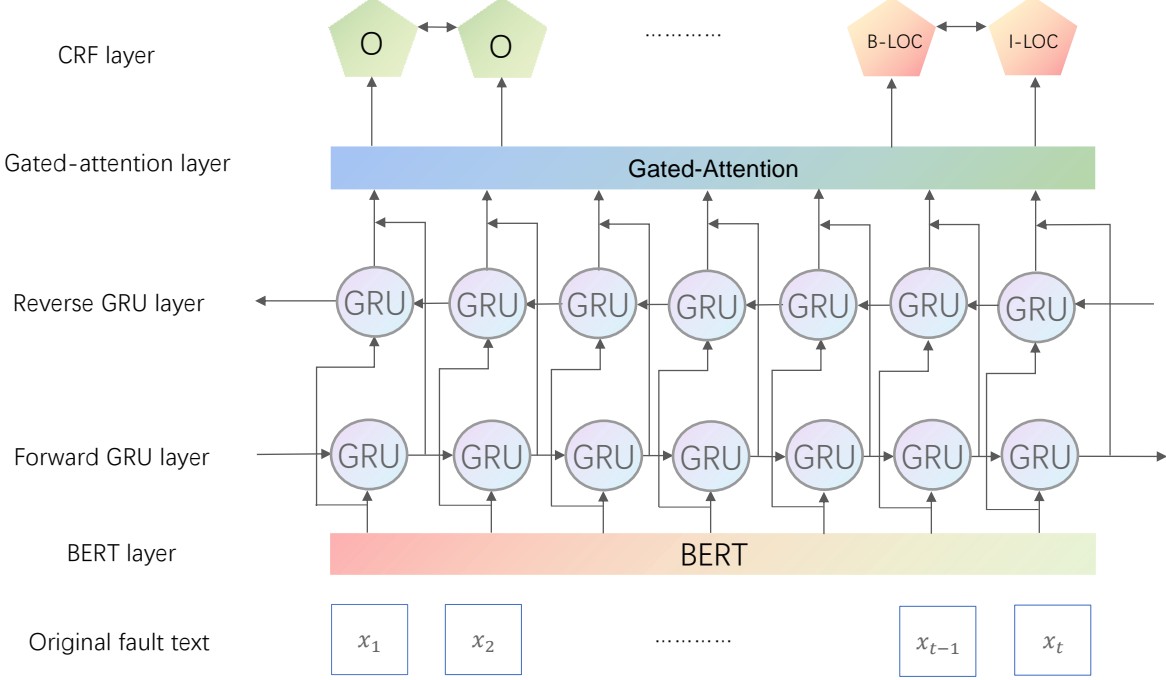

**Figure 5.** BBiGRU-GA-CRF entity extraction model.

The whole entity extraction model is mainly composed of the BERT layer, bidirectional GRU layer, Gated Attention layer, and CRF layer. Next, each module is introduced in detail.

### 3.2.1. BERT Layer

Traditional word embedding methods, such as word2vec, can express the relationship between words, but the relationship between words and vectors is static, which cannot dynamically adjust the representation of the input sequence according to the change in context, which restricts the improvement of entity recognition accuracy to a certain extent. The BERT model introduces a bidirectional transformer coding structure, which can fully understand the semantic information contained in the text, and provides an ideal scheme to solve the above problems [24]. In the proposed BBiGRU-GA-CRF model, the BERT layer undertakes the task of the first step of processing the input text. The BERT layer can be divided into two parts: input layer and output layer, and transformer coding layer. These two parts are introduced in detail below.

(1) Input and output layer

The input of BERT is the processed power system fault text, and the fault text has continuous semantics. In BERT, firstly, the CLS and SEP symbols are inserted into the input statements. CLS precedes the first sentence, and SEP separates the two input statements of the input power fault text. Secondly, BERT overlays the embedded vector, segmentation vector, and position vector of the character to obtain the final vector representation. The vector representation of each character in the input sequence is the embedded vector of the character; the segmentation vector is used to distinguish which specific sentence

each character belongs to, that is, sentence A or sentence B. If the input sequence is single sentence text, it is represented by EA; the position vector is responsible for encoding the position of each character in the input. The above vectors can be obtained through learning in the training process. The output of the BERT model is expressed as a vector, which is then used as the input of the BiGRU module.

(2)  Bidirectional transformer coding layer

In the BERT model, the structure responsible for encoding the input sequence is a multilayer bidirectional transformer encoder, and its model structure is shown in Figure 6. The input of the model is a character vector, and the value is $E_1$, $E_2$ ....$E_n$, $T_1 T_2$ ....$T_n$ is the output vector of the model. The core of the transformer encoder in the BERT model is a Self-Attention Mechanism. Multi-head self-attention is used to obtain the corresponding feature representation of multiple subspaces. Through each self-attention head, the relationship between any two characters is directly calculated. Therefore, the long-distance interdependent features in the input sequence are easier to capture, so the degree of association between these characters can be obtained. Then, the weight of each word is dynamically adjusted according to the correlation between characters, and a new vector representation corresponding to each word is obtained. The new vector representation of this word contains the meaning of the character itself and its relationship with other characters in the sequence. Compared with a single word vector representation, the content is richer and more accurate [25].

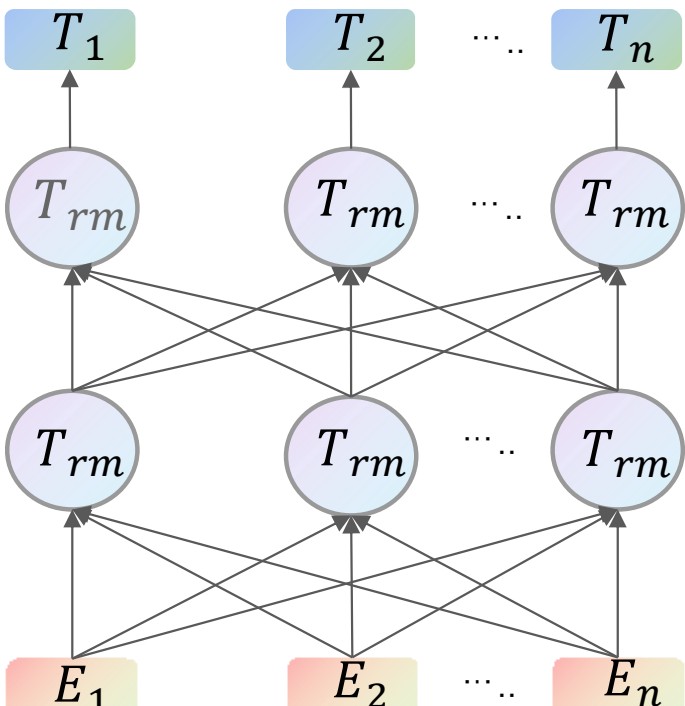

**Figure 6.** BERT multilayer bidirectional transformer encoder.

### 3.2.2. BiGRU Layer

Previous works have used BiLSTM networks, however this paper uses a BiGRU network which has simpler parameters, faster training speed, and an equivalent effect in extracting the global features. The input of the BiGRU module is each vector representation of the output of the BERT model.

The calculation method of each state in the unit is as shown in Equations (16)–(19):

$$r_t = \sigma(w_r C_t + h_{t-1} U_r) \tag{16}$$

$$z_t = \sigma(w_z C_t + h_{t-1} U_z) \tag{17}$$

$$\widetilde{h_t} = tanh(w_h C_t + U_h[r_t \odot h_{t-1}]) \tag{18}$$

$$h_t = (1 - z_t) \odot \widetilde{h_t} + h_{t-1} \odot z_t \tag{19}$$

In the above formula: $z_t$ is the update gate, $C_t$ represents the input vector at time $t$, $r_t$ is the reset gate, $\widetilde{h_t}$ Indicates the candidate hidden state, $h_t$ represents the output of the time $t$ unit. $\odot$ represents the multiplication of corresponding position elements; $w_z$, $w_r$, $w_h$ and $U_z$, $U_r$, $U_h$ are Z respective $z_t$, $r_t$, and $\widetilde{h_t}$. In the training process, the weight matrix is continuously updated.

The BiGRU network is composed of a forward GRU network and backwards GRU network. The input sequence is sent to the two-layer GRU network with opposite direction for calculation. The two-layer GRU network is used to capture historical information and subsequent information. Finally, the output of the two-layer network is spliced according to their respective positions, and the result is the output of the final BiGRU network.

### 3.2.3. Gated-Attention Layer

Due to the problems of traditional Attention Mechanism networks a Gated Attention Mechanism is used for the attention layer. In the Gated Attention layer, the output of the BiGRU layer dynamically selects the matrix elements involved in the calculation after passing through the backbone networks. The elements involved in the calculation are provided corresponding weights, and the eliminated elements are not provided weights. The model is allowed to focus only on the important part of the sequence so as to improve the recognition accuracy of the model.

After obtaining the processed characteristic matrix, this paper used Equation (20) to calculate attention:

$$\text{Attention}(Q, K, V) = \text{SoftMax}(\frac{QK^T}{\sqrt{d_k}})V \tag{20}$$

In this formula, the dot product of the state matrix and its corresponding weight matrix is calculated to obtain the query matrix Q, the key matrix K, and the value matrix V. Among them, the output of the BiGRU layer is used as the state matrix, and the weight matrix is obtained through random initialization. The dimensions of matrices Q and K are $d_k$. The calculation process for attention score is as follows: matrix multiplication is used to multiply Q and K then divided by $\sqrt{d_k}$ to reduce the multiplication result. Finally, softmax is used to normalize the operation to generate a probability distribution and multiply it with matrix V to obtain the final result.

### 3.2.4. Conditional Random Fields (CRF) Layer

Where $m$ is the transfer matrix, the scale is $(k + 2)^2$, and the label of the ith word is $y_i$. Probability using $P_{i,y_i}$ indicates that the size is $(n \times k)$, where $n$ is the sequence length and $k$ is the number of tags. Then the probability of prediction sequence $y$ is as shown in Equation (21):

$$P(Y|X) = \frac{e^{S(X,Y)}}{\sum_{\widetilde{Y} \in Y_X} e^{S(X,\widetilde{Y})}} \tag{21}$$

where $\widetilde{Y}$ Represents the sequence of real markers, and $Y_X$ represents all possible dimension sequences. We used the maximum likelihood training model to ensure that the correct label is obtained with the maximum probability, as shown in Equation (22):

$$\text{LogP}(Y|X) = S(X,Y) - \sum_{\widetilde{Y} \in Y_X} S(X,Y) \tag{22}$$

### 3.3. BiGRU-GA Power Fault Relation Extraction Model

As one of the key steps of constructing a knowledge graph, relationship extraction has an important impact on the quality and availability of the constructed knowledge graph. Traditional methods complete relationship extraction based on pattern matching. For example, Rink et al. [26] proposed a method of relationship extraction by combining semantic relationship and vocabulary. Kambhatl et al. [27] applied the idea of logical regression to relationship extraction. However, the above traditional relationship extraction methods increase the calculation cost and spread errors, resulting in poor universality. Based on the above problems, Huang et al. [28] cleverly combined the cyclic neural network (GRU) and Attention Mechanism, and achieved good results in entity relationship extraction. The model convergence speed in this method is fast, but the fitting effect is general, and the accuracy of relationship extraction needs to be improved.

To solve the above problems, this paper proposes an entity relationship extraction model: Bidirectional Gated Recurrent Unit with Gated Attention Mechanism (BiGRU-GA), which combines BiGRU and an improved Gating Attention Mechanism. According to the data set built in this paper, three types of relationships can be extracted: fault cause, fault phenomenon, and fault solution. In this model, the preprocessed power system fault knowledge corpus is vectorized by word embedding, and the obtained vectorized words are used as the input of the BiGRU two-way gating cycle unit. Through the processing of the BiGRU two-way gating cycle unit, the context semantic features are captured. At the same time, combined with the improved gating Attention Mechanism, the relevant features are further extracted. Finally, the softmax function is used for classification to obtain the optimal result.

The structure of the improved BiGRU-GA relationship extraction model constructed in this paper is shown in Figure 7. The model combines a BiGRU neural network and Attention Mechanism, improves the Attention Mechanism, and adds a gating mechanism to extract the entity relationship in the power system fault knowledge corpus. Firstly, the word vector is used to construct the dictionary. The first 4996 words of the dictionary are selected in this paper; the forward and reverse GRU neural networks are used to encode the power system fault knowledge text. We defined the maximum input length as 128 and construct the fault text token, entity label, and relationship label. The gating Attention Mechanism is constructed on the feature vector to better extract the relationship features of the fault text, eliminate the interference of noise data on the relationship extraction results, and extract the high weight vector. Finally, the relationship between fault entities is extracted by the forward and reverse GRU neural network and softmax classifier. The model proposed in this paper has six layers: the first layer is the input layer, which takes the data in the self-built power system fault knowledge corpus as the input data of the input layer. The second layer is the embedding layer, which embeds the output data of the input layer at the word vector level. The third layer is the forward GRU layer, which encodes the fault statements in the forward direction, and the fourth layer is the reverse GRU layer, which encodes the fault statements in the reverse direction. The fifth layer is the gating attention layer, which improves the Attention Mechanism and introduces the gating mechanism to provide a certain weight to the detected sequence in the fault sentence, rather than to all sequences, which is conducive to further extracting the characteristics of the fault sentence, eliminating the negative impact of noise data, and improving the effect of model relationship extraction. The last layer is the output layer, which outputs the fault relationship type of power system through the joint action of forward and reverse GRU neural network and sigmoid activation function.

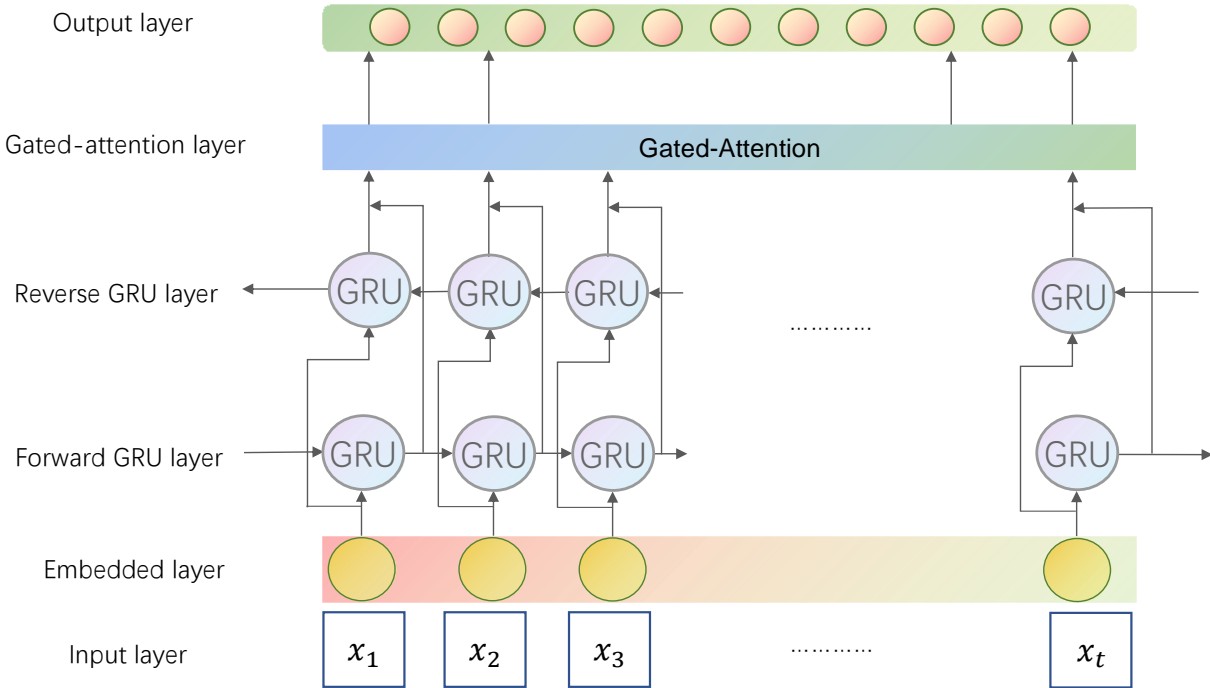

**Figure 7.** BiGRU-GA relation extraction model.

*3.4. Knowledge Graph Construction and Application*

The power fault knowledge graph is constructed using the model proposed earlier, taking the power system fault diagnosis manual as the reference standard, obtaining fault data through the analysis of system logs, and then extracting the entities, the relationship, and the attribute, forming a triplet, and then carrying out knowledge fusion, knowledge processing and other operations, and finally forming the power fault knowledge graph. The build process is shown in Figure 8.

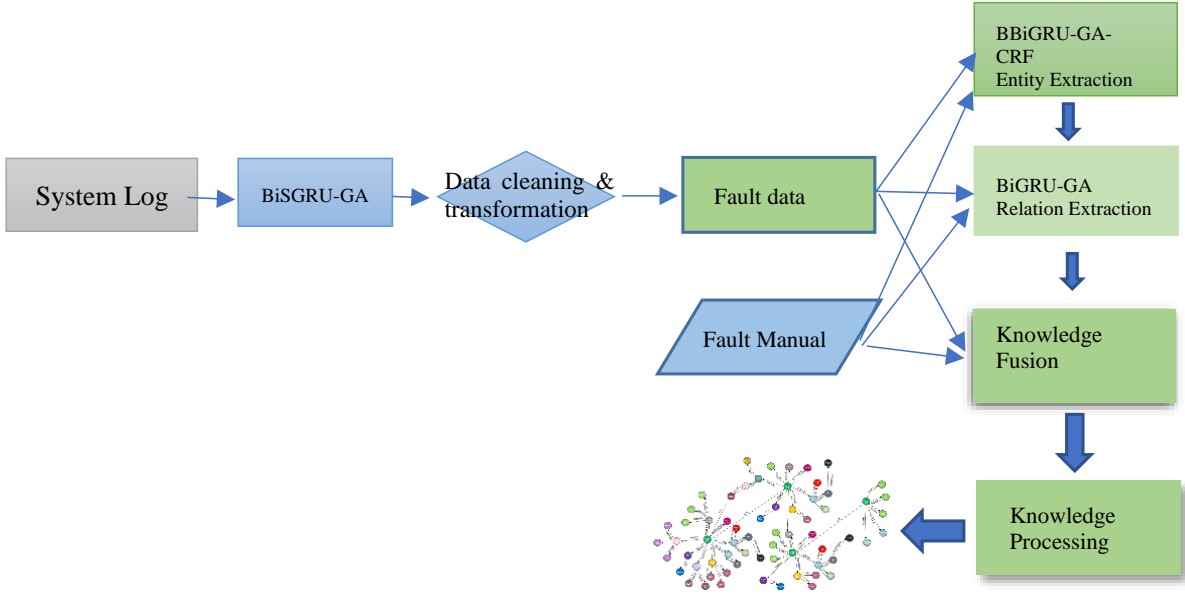

**Figure 8.** Knowledge graph creation process.

We used this method to model the data of a power company in China, forming a total of 2431 nodes and 8947 entity relationships, and importing these data into the Neo4j graph database in the form of entity-relationship-entity triplets, processing and finally obtaining the power system fault knowledge graph as shown in the Figure 9.

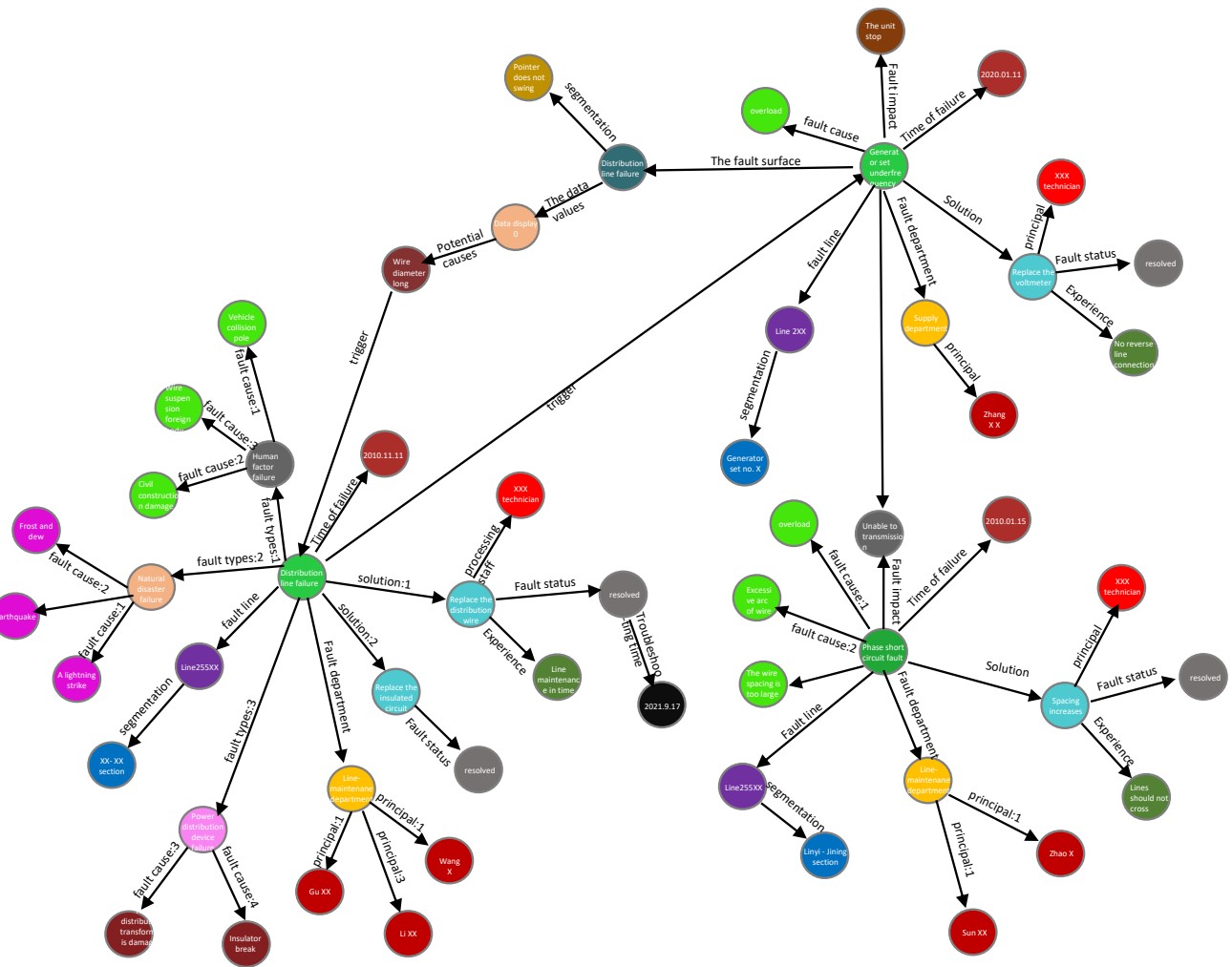

**Figure 9.** Power system fault diagnosis knowledge graph (part).

When the power system fails the supervisory control and data acquisition system (SCADA) collects the fault information (such as fault phenomena, abnormal indicators, fault impact, etc.), analyzes the collected fault information, obtains a detailed fault report, and uses the relevant model of this article to analyze the report content into entity-relationship-entity form of triplet information. The corresponding triplet information is obtained, and then the Cypher query statement in the graph database Neo4j is used to find the matching triple information. If present, it returns the type of failure and the corresponding disposition information. If the matching triplet information is not located in the graph, a human expert analyzes the fault type, combines the power system fault disposal manual, manually processes and organizes the relevant empirical measures to form the corresponding triple, and stores it in the graph database Neo4j. The flow of applying the knowledge graph for troubleshooting is shown in the Figure 10.

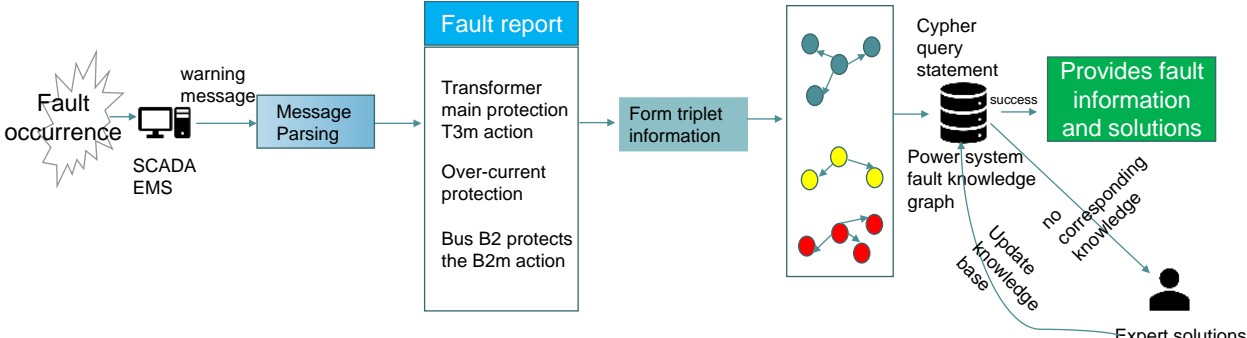

**Figure 10.** Application process of knowledge graph.

## 4. Experiments and Results

In the experimental part, the three models proposed in this paper were verified by experiments. Experiment 1 verified the feasibility and effectiveness of the BiSGRU-GA model. Experiment 2 verified the feasibility and effectiveness of the BBiGRU-GA-CRF model. Experiment 3 verified the feasibility and effectiveness of the BiGRU-GA model.

### 4.1. Experiment 1–BiSGRU-GA

#### 4.1.1. Data Set and Data Preprocessing

The data set selected in this experiment is from the operation log of the all-business Unified Data Center of the Shandong Electric Power company of China, with a total of 26,433 log messages. Each operation log information includes time (year, month, day, hour, minute, and second), source IP, data size, and other fields, and has a unique data block ID. The power operation and maintenance expert divides the operation log into 5647 event samples according to the data block ID and log information. Among them, about 0.8% of 45 samples were marked as abnormal. Therefore, we take the samples being labeled abnormal or not as the basis for the evaluation of detection accuracy. Several rounds of comparative tests are carried out to evaluate the performance of the model in this paper.

#### 4.1.2. Experimental Environment

The experimental environment is as follows: the model implementation and related improvements are in the open-source framework PyTorch 1.9.1. The computer CPU model used in the training is an Intel (R) Xeon (R) platinum 8124 M, the GPU model is a GeForce RTX 3090, the size of video memory is 24 GB, and the operating system used is CentOS7.

#### 4.1.3. Evaluation Index

The evaluation indexes used in the experiment include accuracy, precision, recall, F-measure, false positives (FPs), and false negatives (FNs). F-measure is a comprehensive evaluation index based on accuracy and recall. FP represents the number of false positives in the exception log, that is, it is actually abnormal and predicted to be normal. *FN* represents the number of false positives in the normal log, that is, it is actually normal and predicted to be abnormal. TP represents the correct predicted number of normal logs, i.e., true positive. *TN* represents the number of abnormal logs correctly predicted, i.e., true negative. Accuracy indicates the percentage of the number of normal logs and abnormal logs accurately classified in the total number of logs; the accuracy calculation formula is as shown in Equation (23):

$$Accuracy = \frac{TP + TN}{TP + TN + FP + FN} \tag{23}$$

Precision indicates the proportion of the number of detected true exception logs in the number of all detected exception logs, the calculation formula of precision is as shown in Equation (24):

$$Precision = \frac{TP}{TP + FP} \tag{24}$$

Recall indicates the percentage of the number of detected exception logs in the total number of exception logs in the data set, the calculation formula of the recall rate is as shown in Equation (25):

$$Recall = \frac{TP}{TP + FN} \tag{25}$$

The calculation formula of F-measure is as shown in Equation (26):

$$F\text{-}measure = \frac{2 * Precision * Recall}{Precision + Recall} \tag{26}$$

### 4.1.4. Model Parameters

The word embedding dimension is set to 200, the batch_size is set to 64, the number of hidden layer nodes is set to 100, and the optimizer is Adam. We chose 100 iterations. The model parameters are shown in Table 1:

**Table 1.** BiSGRU-GA Model parameter setting.

| Parameter Name | Parameter Value |
| :---: | :---: |
| Word vector dimension | 200 |
| Batch_size | 64 |
| Number of hidden nodes | 100 |
| Optimizer | Adam |
| Dropout | 0.5 |
| Number of iterations | 100 |

### 4.1.5. Experimental Results

We selected three widely used anomaly detection models to compare with the proposed model: PCA, DeepLog, and a GRU-based Deep Learning Anomaly Detection Model. PCA is a model for log anomaly detection based on offline mode, which was implemented by He S et al. [29]. DeepLog constructs an anomaly detection model based on LSTM, which is reproduced in this paper based on [22]. The GRU-based Deep Learning anomaly detection model is an improvement on the basic DeepLog model and can complete the online detection of log anomalies. Firstly, the experiment compares the detection effects of the PCA, DeepLog and GRU models, and then compares the detection speed of the PCA, DeepLog, and GRU models with the BiSGRU-GA model proposed in this paper, so as to comprehensively evaluate this model from many aspects.

The comparative experiment of this paper uses the operation log data set of the whole business unified data center of a provincial power company. We divided the 5647 log event samples (i.e., normal or daily logs) into a training set and test set according to the ratio of 3:1. Table 2 provides the specific information of the training set and test set used in this experiment.

**Table 2.** Training set and test set data information.

| Number of Normal Logs | Number of Abnormal Logs | Number of Log Keys |
| :---: | :---: | :---: |
| Training set 4235 | 0 | 17 |
| test set 1411 | 45 | 17 |

Table 3 and Figure 11 shows the performance of the four models. It can be seen that the PCA model performs well in false positive indicators, indicating that the PCA model

controls the abnormal false positive rate well, but obtains more false negatives, indicating that the PCA model tends to judge the abnormal logs as normal. In contrast, the DeepLog and GRU models achieved less false positives and false negatives. The BiSGRU-GA model is slightly better than the DeepLog model and the GRU model in terms of false positives and false negatives. The performance is improved to a certain extent.

**Table 3.** Performance comparison of PCA, DeepLog, GRU.

| FP | FN |
|---|---|
| PCA 18 | 64 |
| DeepLog 11 | 35 |
| GRU 9 | 32 |
| BiSGRU-GA 7 | 21 |

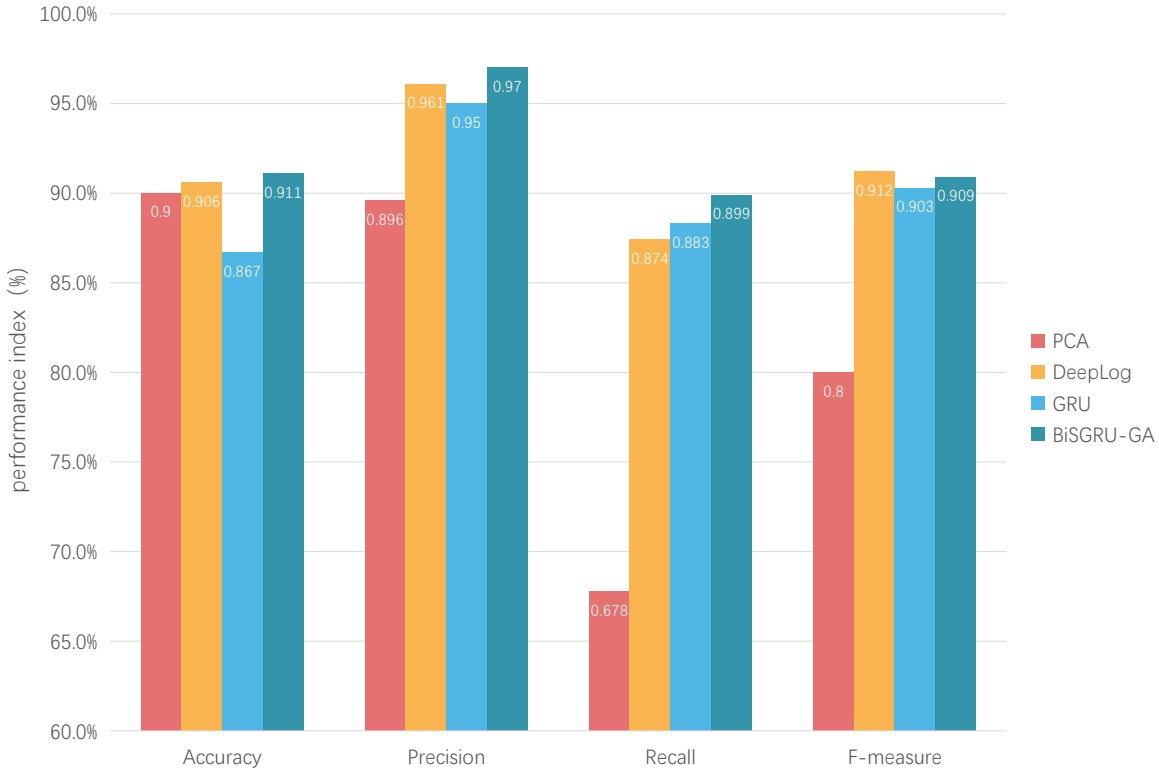

**Figure 11.** Experimental results of anomaly detection model performance comparison.

Figure 11 compares the above four models in terms of accuracy, recall, and F-measure. As can be seen from the figure, except for the F-measure index of the proposed model being slightly lower than that of the DeepLog model, the proposed model achieved the highest results in the three indexes of accuracy, precision, and recall. Compared with the PCA model, this model increased by 1.1%, 7.4%, 22.1%, and 10.9% in the four indexes of accuracy, precision, recall, and F-measure, respectively. Compared with the DeepLog model, the accuracy rate, precision rate, and recall rate increased by 0.5%, 0.9%, and 2.5%, respectively. The F-measure is 0.3% lower, and compared with the GRU model, the accuracy, precision, recall, and F-measure increased by 4.4%, 2.0%, 1.6%, and 0.6%, respectively. The overall performance is slightly better than the DeepLog model and the GRU model, and significantly better than PCA model.

In order to verify the improvement of the running speed of this model, we compared this model with the PCA, DeepLog, and GRU-based deep learning anomaly detection models. Two indicators are used to measure the running speed of the model: the total time spent detecting all logs and the average time spent detecting each log. At the same time, we

defined the time cost of the PCA model with the largest time cost as 1. Table 4 shows the running speed of the GRU model and the BiSGRU-GA model on this paper's log data set:

**Table 4.** Efficiency comparison of PCA, DeepLog, GRU.

| Total Number of Logs | Total Time (s) | Average Time (ms) | Time Cost |
|---|---|---|---|
| PCA 26,433 | 117 | 4.43 | 1 |
| DeepLog 26,433 | 101 | 3.82 | 0.86 |
| GRU 26,433 | 85 | 3.21 | 0.72 |
| BiSGRU-GA 26,433 | 73 | 2.76 | 0.62 |

Combined with Figure 11 and Table 4, it can be seen that on the premise of achieving the best detection effect, the model proposed in this paper has a great lead in running speed. Compared with the PCA model, DeepLog model, and GRU model, the running time is reduced by about 37.6%, 27.7%, and 13.9%, respectively, which reflects the advantages of this model in terms of time overhead.

### 4.1.6. Experimental Conclusion

On the whole, this model is better than the PCA model and the DeepLog model. At the same time, it is better than the GRU model in time cost, and the detection speed is faster. In the face of the huge number of power system logs and high real-time requirements, this model has a high application value and practical significance.

### 4.2. Experiment 2–BBiGRU-GA-CRF
#### 4.2.1. Data Set and Data Preprocessing

The fault text data set used in this paper comes from the power system fault knowledge Manual of a province (hereinafter referred to as the manual). The main components of the manual are the power fault handling manual sorted out by the whole business unified data center of a provincial power company and the fault knowledge manually sorted based on the abnormal log data of the power system. It contains more than 40,000 Chinese entity identification and marking data. The entity categories are divided into four categories: fault name, fault phenomenon, post fault operation mode, and fault handling measures. In this paper, the data set is divided into a training set, verification set, and test set according to the ratio of 3:1:1. The specific scale of the data set is shown in Table 5.

**Table 5.** Specific size of data set for entity extraction.

| Data Set | Training Set | Validation Set | Test Set |
|---|---|---|---|
| Power System Fault Knowledge Manual | 24,162 | 8154 | 8469 |

#### 4.2.2. Experimental Environment

The experimental environment is the same as that of Experiment 1.

#### 4.2.3. Evaluation Index

The evaluation indexes used in the experiment include accuracy, recall, and F-measure. The calculation method is the same as Experiment 1.

#### 4.2.4. Model Parameters

The experimental parameter settings are shown in Table 6.

**Table 6.** BBiGRU-GA-CRF Model parameter setting.

| Parameter Name | Meaning | Parameter Value |
|:---:|:---:|:---:|
| batch_size | Number of samples per batch | 64 |
| max_seq_len | Number of words | 64 |
| lr | Initial learning rate | 0.001 |
| Dropout | Proportion of discarded neurons | 0.5 |
| BERT-Base | Number of hidden layers of BERT base | 12 |

In order to verify the advanced nature of the BBiGRU-GA-CRF entity recognition model in the field of fault text recognition, we selected the following classical entity recognition models for comparative experiments and analyzed the following models:

1. BiLSTM-CRF [30] model: a BiLSTM network is used to capture two-way semantic dependencies, and then capture global features;
2. BiGRU-CRF [31] model: replaces the BiLSTM network with a BiGRU network which uses fewer parameters;
3. BERT-BiLSTM-MHA-CRF [32] model: a Multi-head Attention Mechanism is introduced to capture local features;
4. BBiGRU-GA-CRF model: the model proposed in this paper.

4.2.5. Experimental Results

The above four models are used for comparative experiments on the data set, and the experimental results are shown in the Figures 12 and 13.

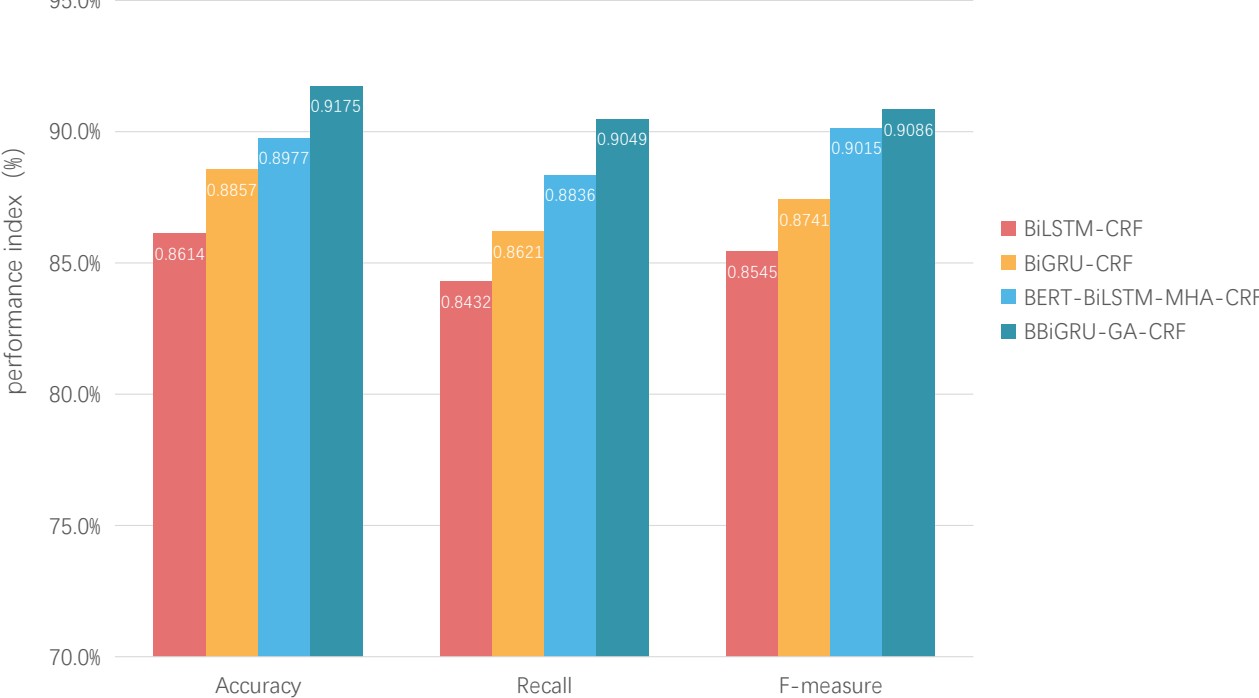

**Figure 12.** Experimental results of entity extraction model performance comparison.

As can be seen from the experimental results in Figures 12 and 13, compared with the BiLSTM-CRF model and the BiGRU-CRF model, the accuracy, recall, and F-measure of the BBiGRU-GA-CRF model improved by 5.61%, 6.17%, and 5.41%, and 3.18%, 4.28% and 3.45%, respectively. However, the time consumption of each round increased by 161% and 829%, respectively. This aptly demonstrates that adding a BERT layer can fully extract character level features, dynamically adjust the vector representation of characters according to the changes in semantic environment, significantly improve the generalization ability of the

model, and significantly enhance the effect of entity recognition; however, it also increases the time overhead. Compared with BERT-BiLSTM-MHA-CRF, the accuracy, recall, and F-measure of this model were improved by 1.98%, 2.13%, and 0.71%, respectively. Moreover, the time cost per round is reduced by 28%. It aptly demonstrates that the introduction of GRU and the Gated Attention Mechanism can capture local features, improve the model recognition effect, and reduce the time cost.

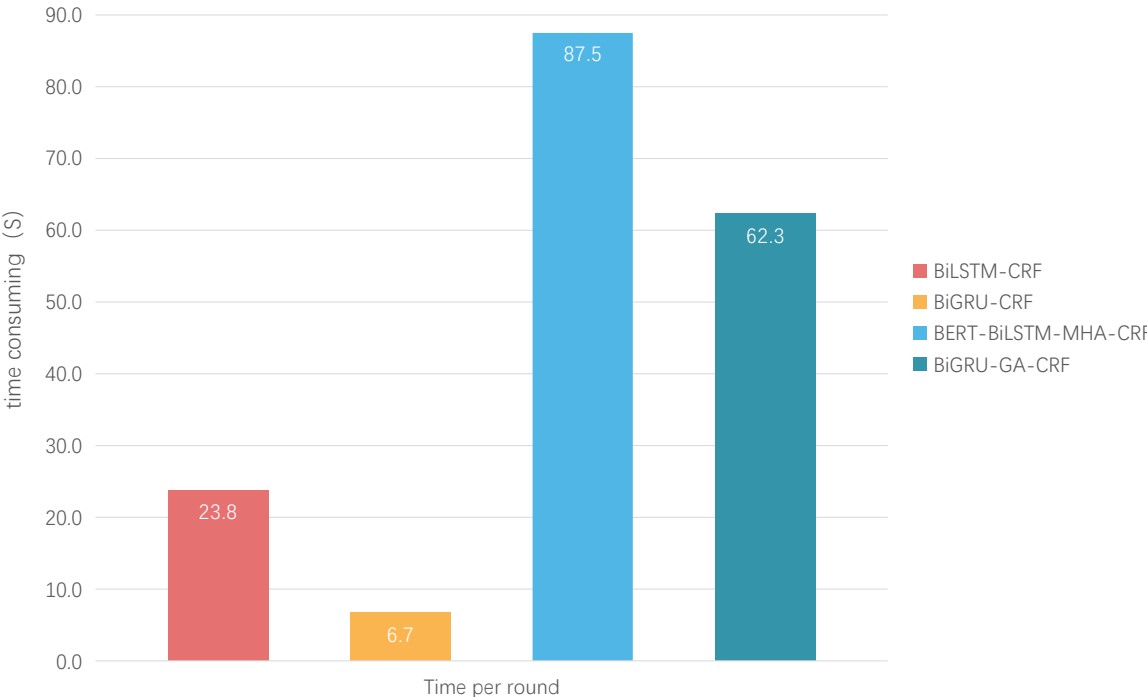

**Figure 13.** Each round of the entity extraction model takes time to compare the experimental results.

In order to verify the effects of the addition of a BERT layer and Gated Attention Mechanism on the experimental results, two groups of ablation experiments were set up in this paper. The model with only the BERT layer was compared with other classical models, and the model with only the Gated Attention Layer was compared with other classical models. The experimental results are shown in the Figures 14 and 15.

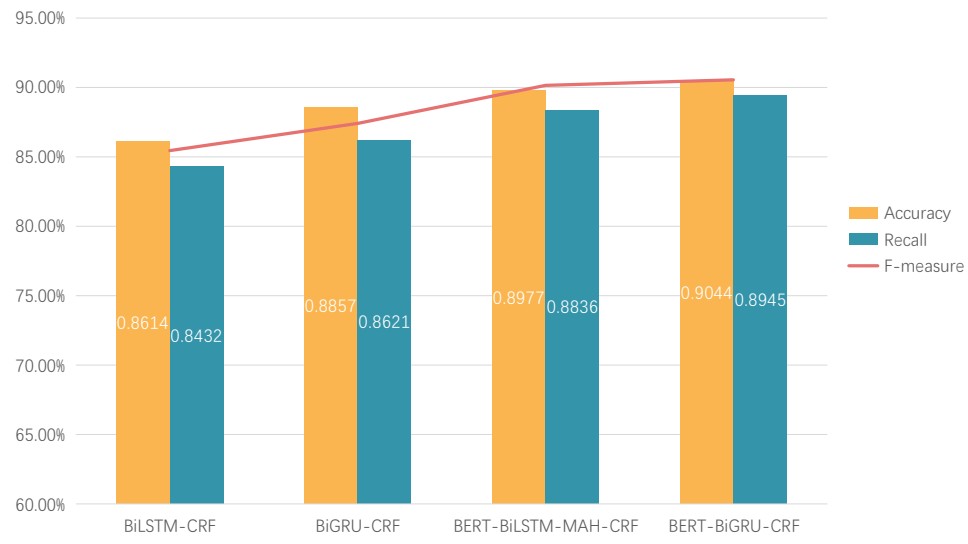

**Figure 14.** Ablation experiment, BERT layer.

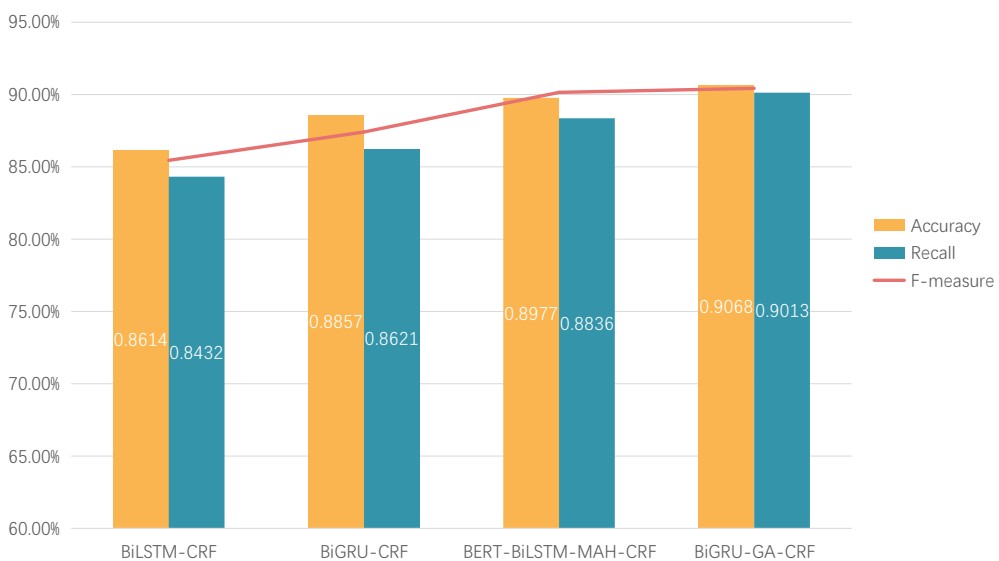

**Figure 15.** Ablation experiment, GA layer.

Different dimension vectors generated by BERT model training also have a certain impact on the experimental results. This experiment used $GRU_{dim}$ values of 100, 200, 300, and 400 for comparison. The comparison results are shown in Figure 16. It can be seen that the F1 value extracted by the entity is the best when the vector dimension is set to 300. When the dimension is low, the vector features are not complete enough, resulting in underfitting. When the dimension is too high, the noise generated in the experiment is captured, resulting in overfitting.

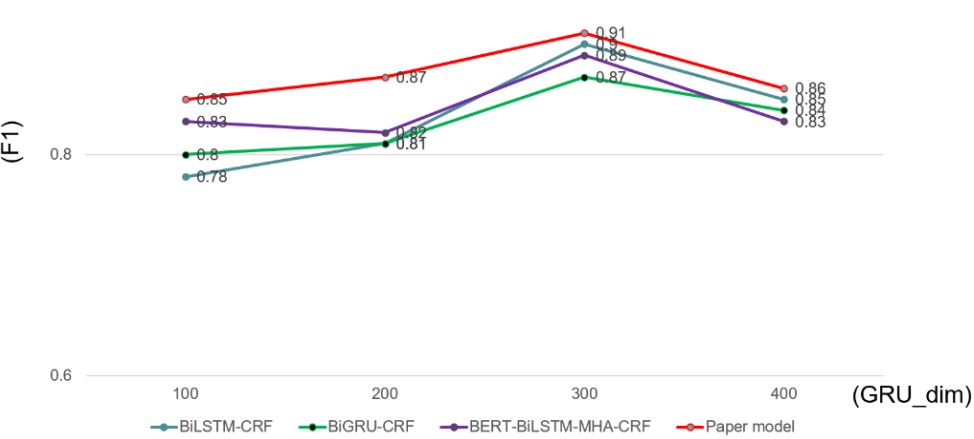

**Figure 16.** Comparison of entity extraction models experimental results in different dimensions.

### 4.2.6. Experimental Conclusion

Through the analysis of the experimental results of the above models, it was proved that the BBiGRU-GA-CRF entity recognition model proposed in this paper not only improves the overall recognition accuracy, but also reduces the time cost of training, and achieves the comprehensive optimal effect.

### 4.3. Experiment 3–BiGRU-GA

### 4.3.1. Data Set and Data Preprocessing

The data sources are the power fault handling manual sorted out by the whole business unified data center of a provincial power company and the fault knowledge manually sorted out based on the abnormal log data of the power system in this paper. It contains more than 40,000 pieces of Chinese entity identification and annotation data. Based on the

self-built data set, this paper arranged a total of 3625 sets of entity pairs that determine the relationship, which were divided into three relationship types: fault causes, fault phenomena, and solutions. In this paper, the data set was divided into training set and test set according to the ratio of 7:3. The specific scale of the data set is shown in Table 7, and the relationship type and quantity are shown in Table 8:

**Table 7.** Specific size of data set for relation extraction.

| Data Set | Training Set | Test Set |
|:---:|:---:|:---:|
| 3625 | 2537 | 1087 |

**Table 8.** Relationship type and quantity.

| Relationship Type | Quantity |
|:---:|:---:|
| Cause of failure | 1627 |
| Fault phenomenon | 542 |
| Solutions | 1456 |

### 4.3.2. Experimental Environment

The experimental environment is the same as that of Experiment 1.

### 4.3.3. Evaluation Index

The evaluation indexes used in the experiment include accuracy, recall, and F-measure. The calculation method is as shown in Equations (27)–(29):

$$Accuracy = \frac{\sum_1^n P_i}{n} = \frac{\sum_i^n \frac{T_i}{\dot{I}}}{n} \times 100\% \tag{27}$$

$$Recall = \frac{\sum_1^n P_i}{n} = \frac{\sum_i^n \frac{T_i}{I}}{n} \times 100\% \tag{28}$$

$$F\text{-}measure = \frac{2PR}{P+R} \times 100\% \tag{29}$$

where $T_i$ represents the number accurately predicted as class $i$, $I$ represents the actual number of class $i$, and $\dot{I}$ represents the predicted number of class $i$, and $n$ is the total number of relationship types.

### 4.3.4. Model Parameters

The experimental parameter settings are shown in Table 9:

**Table 9.** Relation extraction experimental parameters.

| Parameter Name | Meaning | Parameter Value |
|:---:|:---:|:---:|
| batch_size | Number of samples per batch | 16 |
| lr | Initial learning rate | 0.001 |
| num_class | Number of relationship categories | 3 |
| num_epoch | Number of iterations | 64 |

In order to verify the advanced nature of the BiGRU-GA relation extraction model in the field of fault text relation extraction, we selected the following classical entity recognition models for comparative experiments and analyzed the following models:

1.  BiLSTM [33] model: a BiLSTM network is used to capture two-way dependence, and then capture global features;
2.  BiGRU [34] model: replaces the BiLSTM network with a BiGRU network which uses fewer parameters;

3. BiLSTM attention [35] model: an Attention Mechanism is introduced to better extract features from the BiLSTM model;

4. BiGRU-GA model: based on BiGRU attention model, the improved model proposed in this paper introduces a Gating Attention Mechanism to strengthen feature extraction and enhance the recognition effect of the model.

### 4.3.5. Experimental Results

The above four models are used for comparative experiments on the data set, and the experimental results are shown in the Figures 17 and 18:

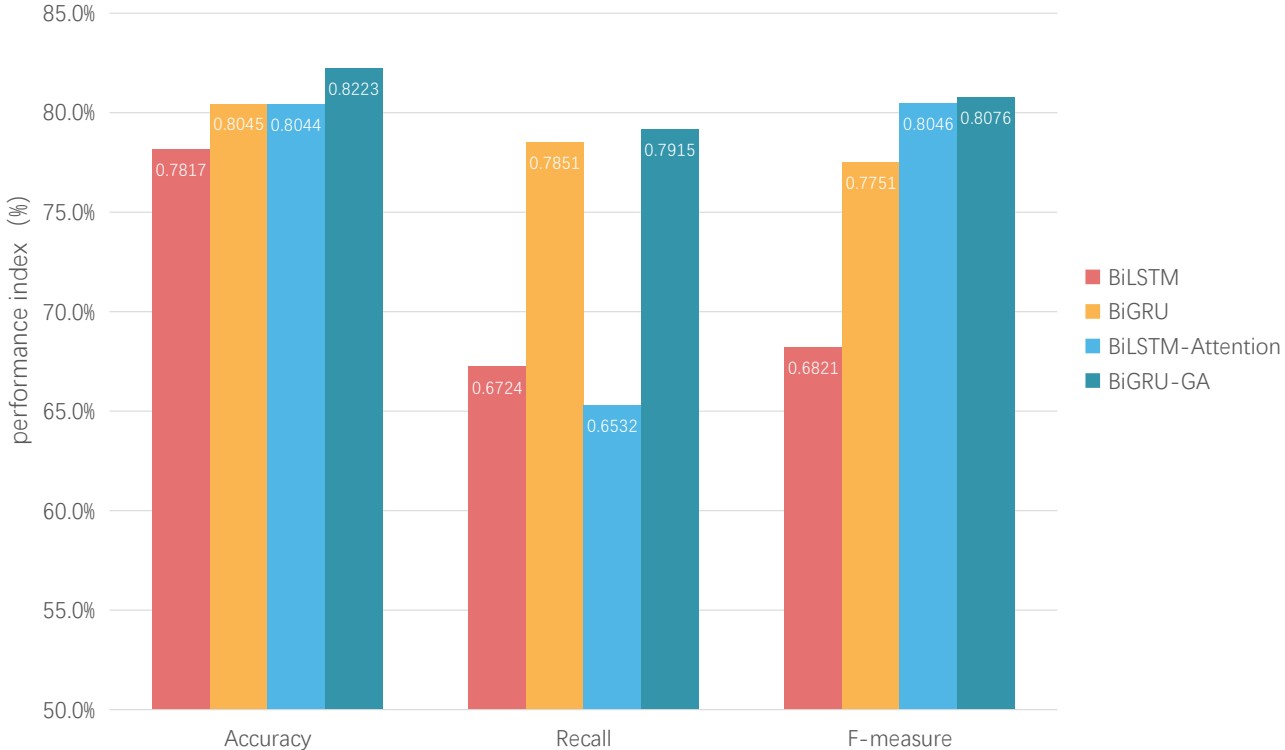

**Figure 17.** Experimental relation extraction results of model performance comparison.

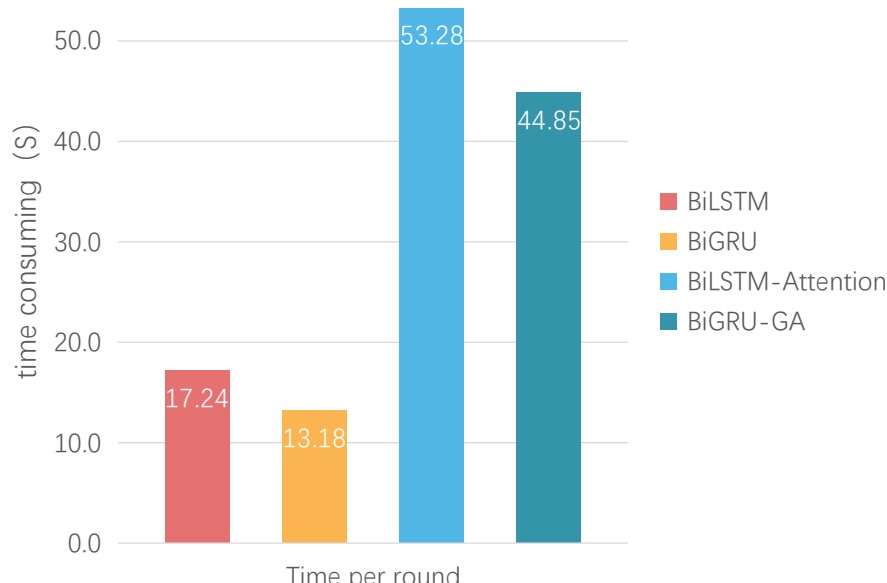

**Figure 18.** Each round of the relation extraction model takes time to compare the experimental results.

As can be seen from the experimental results in Figures 17 and 18, compared with the BiLSTM model, the accuracy, recall, and F-measure of this model were improved by 4.66%, 11.91%, and 12.55%, respectively. Compared with the BiGRU model, the accuracy, recall and F-measure of this model were improved by 1.78%, 0.64%, and 3.25% respectively. Compared with the BiLSTM Attention model, the accuracy, recall, and F-measure of this model were improved by 1.79%, 13.83%, and 0.30%, respectively, and the time overhead was reduced by about 16%. The results aptly demonstrate that the introduction of a two-way GRU neural network with fewer parameters and a Gated Attention Mechanism can capture local features and reduce the cost of training time, whilst also improving the effect of model recognition.

Different dimension vectors generated by BERT model training also have a certain impact on the experimental results. This experiment will use $GRU_{dim}$ values of 100, 200, 300, and 400 for comparison. The comparison results are shown in Figure 19. It can be seen that the F1 value extracted by the entity is the best when the vector dimension is set to 300. When the dimension is low, the vector features are not complete enough, resulting in underfitting. When the dimension is too high, the noise generated in the experiment will be captured, resulting in overfitting.

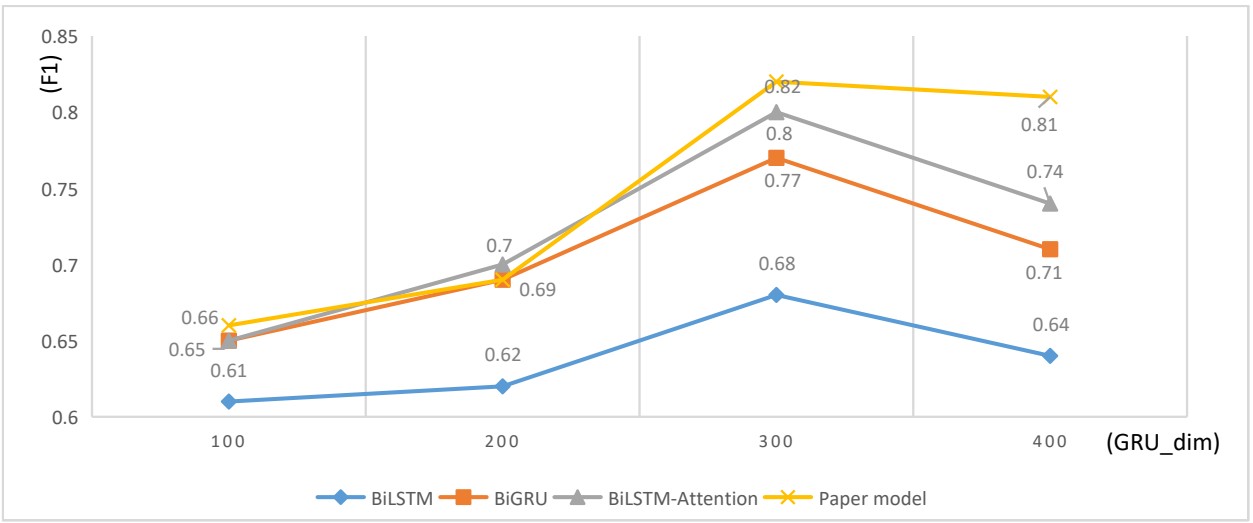

**Figure 19.** Comparison of relation extraction models experimental results in different dimensions.

4.3.6. Experimental Conclusions

Through the analysis of the experimental results of the above models, it was proved that the BiGRU-GA relationship extraction model proposed in this paper not only improves the overall recognition accuracy, but also reduces the training time overhead and achieves the comprehensive optimal effect.

**5. Conclusions**

The main work of this paper is summarized as follows:

1.  Aiming at the problems in the process of constructing a fault knowledge graph, this paper proposed a log anomaly detection model: BiSGRU-GA, which can quickly and efficiently detect the anomaly log in the power system and serve as fault data after further analysis and processing, it enriches the source of fault data and facilitate relationship analysis;
2.  Aiming to solve the problems of the low accuracy of traditional entity recognition models and relationship recognition models and being unable to make full use of the context information of sentences in the construction of fault knowledge graph, an entity extraction model based on BBiGRU-GA-CRF and a relationship extraction model based on BiGRU-GA are proposed. On the basis of making full use of the

context information of sentences, higher accuracy of entity extraction and relationship extraction is achieved.

Based on knowledge graph technology and system log anomaly detection technology, this paper makes a detailed study on the construction of a power system fault knowledge graph and its application in fault diagnosis. However, due to the limitations of the researchers' knowledge and practical experience in this field, more in-depth research and practice are still needed for entity and relationship extraction methods and fault diagnosis algorithms. Future research work will be implemented from the following aspects:

1. In terms of fault data, although this paper obtains a large number of fault data based on the power system anomaly log and expands the source of fault data, the scale of the data set is still small. In the future research work, we will continue to collect relevant fault data to expand the scale of the data set. In addition, we should continue to deeply study the ontology construction technology to enhance the preciseness of fault data, so as to better complete the research work of entity and relationship extraction;

2. In the theoretical research of models and methods, the BiSGRU-GA model, BBiGRU-GA-CRF model, and BiGRU-GA model used in this paper improved on the basis of the mainstream deep learning model, and achieved good results. In future study and research work, we will try to apply the above model to other relevant production fields, and improve and optimize the problems and shortcomings exposed by the model according to the subsequent experimental results.

**Author Contributions:** Conceptualization: P.L. and B.T.; methodology: P.L. and S.G.; resources: X.L., L.Y. and Y.L.; data curation: C.M. and W.Z.; writing—original draft preparation: S.G.; writing—review and editing: P.L. and L.B.; supervision: P.L.; funding acquisition: S.G. All authors have read and agreed to the published version of the manuscript.

**Funding:** This work is supported by China's national key R&D plan and supported by the project name: research and application demonstration of key technologies of integrated emergency communication based on radio and television system, subject No. 2017YFC0806200.

**Institutional Review Board Statement:** Not applicable.

**Informed Consent Statement:** Not applicable.

**Acknowledgments:** We are grateful to the anonymous reviewers for comments on the original manuscript.

**Conflicts of Interest:** The authors declare no conflict of interest.

## Abbreviations

| Nomenclature | |
|---|---|
| GA | Gated Attention Mechanism |
| CRF | Conditional Random Fields |
| MHA | Multi-Headed Attention Mechanism |
| GRU | Gated Recurrent Units |
| LSTM | Long Short-Term Memory |
| BERT | Bidirectional Encoder Representations from Transformers |
| BiGRU | Bidirectional GRU |
| BiSGRU | Bidirectional Sliced GRU |
| BiGRU-GA | BiGRU with Gated Attention Mechanism |
| BiSGRU-GA | BiSGRU with Gated Attention Mechanism |
| BiLSTM | Bidirectional LSTM |

| | |
|---|---|
| BiLSTM-Attention | BiLSTM with standard Attention Mechanism |
| BiLSTM-CRF | BiLSTM with Conditional Random Fields |
| BERT–BiLSTM–CRF | BiLSTM-CRF with BERT input layer |
| BiGRU-CRF | BiGRU with Conditional Random Fields |
| BiGRU-GA-CRF | BiGRU-CRF with Gated Attention Mechanism |
| BBiGRU-GA-CRF | BiGRU-GA-CRF with BERT input layer |
| BERT-BiLSTM-MHA-CRF | BERT–BiLSTM–CRF with a multi-headed Attention Mechanism |
| List of Symbols | |
| $z_t$ | Update door, the degree of influence the previous state transition has on the current state |
| $r_t$ | Reset door, controls the extent to which information from the previous state is written to the previous candidate set $\widetilde{h}_t$ |
| $h_t$ | Hidden state of current node |
| $\widetilde{h}_t$ | Hidden state after $r_t$ processing |
| $y_t$ | Output of output layer |
| $X_i$ | $X_i$ stands for the ith log key |
| $\overrightarrow{h_{ij}^o}$ | Vectorization status after GRU processing in the forwards direction |
| $\overleftarrow{h_{ij}^o}$ | Vectorization status after GRU processing in the backwards direction |
| $\overrightarrow{h_t^1}$ | The result obtained after processing the output sequence of theattention layer in the forward direction |
| $\overleftarrow{h_t^1}$ | The result obtained after processing the output sequence of the attention layer in the backward direction |
| $h_t^1$ | Represents the implicit representation of the t-th log key subsequence of the first layer |
| Y | Output results of BiSGRU-GA |
| $\overrightarrow{Y}$ | Forward GRU output results |
| $\overleftarrow{Y}$ | Backward GRU output results |
| LogP(Y|X) | The maximum likelihood training model takes the maximum probability result of Y as the final judgment result of the model. |
| p($x_t$|C) | Probability of occurrence of each log key Attention mechanism weight |
| P(Y|X) | Probability of generation of sequence y |
| *Accuracy* | The proportion of correct forecast and classification to the total data volume, i.e., accuracy |
| *Precision* | Percentage of the number of normal logs and abnormal logs accurately classified in the total number of logs |
| *Recall* | Indicates the proportion of the number of detected true exception logs in the number of all detected exception logs or the proportion of correctly classified entries in the total classification entries |
| *F − measure* | Weighted harmonic average of *Precision* and *Recall* |

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
