# Peer review of "Construction of Power Fault Knowledge Graph Based on Deep Learning"

_applsci, doi:10.3390/app12146993_

Round 1

Reviewer 1 Report

The idea is interesting, but the following concerns should be considered.

The quality of the images is terrible. The structure of references should be modified. Some of them are not complete. The abstract uses several undefined acronyms! The abstract is ok but it seems that it can be rewritten for the sake of simplicity. the structure of the paper should be added at the end of the introduction. many subsections are used without explanations about each section. This problem in section 3 must be solved and an explanation of big-picture about the method before opening the section must be given. The idea behind of design of the experiment should be added in the part of the experimental results. The paper has no conclusion section!!!

Author Response

Dear Professor,
Thanks for your suggestions, we have modified the paper as follows:
1. The image is redrawnï¼›
2. The references were modified, but some of them came from the Internet and lacked page numbersï¼›
3. Modify the abstractï¼›
4. Acronyms are explained intensivelyï¼›
5. The structure of the paper is addedï¼›
6. Added a section descriptionï¼›
7. The idea of experimental design is providedï¼›
8. The conclusion of the paper is providedï¼›
Now we provide our revised manuscript for your guidance.
thank you

Reviewer 2 Report

The authors proposed an approach to effective recognition fault sentences.  Authors have proposed the BBiGRU-GA-CRF for designing the proposed approach. The approach is seems to be simple and effective, however, the structure of the manuscript can be improved for better showcase. My main concerns about the manuscript are:

1.      A concise and factual abstract is required to state briefly the purpose of the research, the principal results and the major conclusions. Line number 13-28 can be shorten and pin-pointed.

2.      Full form of all Abbreviations should be properly written when it is initially called in the text. My suggestion is to add a separate Nomenclature section.

3.      Literature review is too weak. Add little more related work in the literature review.

4.      I suggest, devide the introduction section in to (i) Motivation and incitement (ii) Literature review and Research gaps (iii) Major Contribution and Organization

5.      Authors should justify their contribution towards novelty. Major Contributions should be presented in bullet notes.

6.      Provide a flow chat for the proposed method.

7.      As this work uses Deep learning for Power system studies, the following references may be useful for the literature study. https://doi.org/10.1002/er.5331, https://doi.org/10.1016/j.engappai.2020.104000, https://doi.org/10.17775/CSEEJPES.2020.02700

8.      The dataset used for the testing of the proposed approach is standardised? Justify.

9.      Not a single figure is visible clearly. Specifically, in the results section, It is so difficult to analyse.

10.  Conclusion section is missing.

Author Response

Dear Professor,
Thanks for your suggestions, we have modified the paper as follows:
1. The image is redrawnï¼›
2. The references were modified, but some of them came from the Internet and lacked page numbersï¼›
3. Modify the abstractï¼›
4. Acronyms are explained intensivelyï¼›
5. The structure of the paper is addedï¼›
6. Added a section descriptionï¼›
7. The literature review section is supplemented
8. The idea of experimental design is providedï¼›
9. The conclusion of the paper is providedï¼›
Now we provide our revised manuscript for your guidance.
thank you

Round 2

Reviewer 1 Report

the authors did reasonable work and many of my concerns have been resolved but the flowing problem from the previous review has non been addressed yet.

"many subsections are used without explanations about each section."

in addition,

1. the symbols should be explained in more detail

2. more related works can be added to the paper

Author Response

We gratefully appreciate for your valuable suggestion, below the comments are response point by point:

1. many subsections are used without explanations about each section

 Reply: we add explanation for each section;

2. the symbols should be explained in more detail

Reply: we add a symbol section to explain the symbol.

3. more related works can be added to the paper

Reply: we add more literature review in introduction section.

Reviewer 2 Report

Authors have improved the paper significantly especially the figures part. However, as the authors have not highlighted the changes they have done, It is difficult to analyse the paper. Few are the comments need to be responded by the authors such as

     1. My suggestion is to add a separate Nomenclature section.

3   2. Literature review is too weak. Add little more related work in the literature review. (Please highlight the changes)

4 3. I suggest, devide the introduction section in to (i) Motivation and incitement (ii) Literature review and Research gaps (iii) Major Contribution and Organization

5 4. Authors should justify their contribution towards novelty. Major Contributions should be presented in bullet notes.

5   5. Provide a flow chat for the proposed method.

7  6. As this work uses Deep learning for Power system studies, the following references may be useful for the literature study. https://doi.org/10.1002/er.5331, https://doi.org/10.1016/j.engappai.2020.104000, https://doi.org/10.17775/CSEEJPES.2020.02700

Author Response

We gratefully appreciate for your valuable suggestion. Each suggested revision and comment , brought forward by the reviewers was accurately incorporated and considered. Below the comments are response point by point and the revisions are indicated.

1. My suggestion is to add a separate Nomenclature section.

Reply: we add a separate Symblo section to  explain the symbol.

2. Literature review is too weak. Add little more related work in the literature review. (Please highlight the changes)

Reply:  we add more literature review in introduction section.

3. I suggest, devide the introduction section in to (i) Motivation and incitement (ii) Literature review and Research gaps (iii) Major Contribution and Organization

Reply: The introduction has been revised, The contents include (I) Motivation and incitement (II) Literature review and Research gaps (III) Major Contribution and Organization parts, but considering that few papers are divided into sections in the introduction part, we do not have a separate section.

4. Authors should justify their contribution towards novelty. Major Contributions should be presented in bullet notes.

Reply: we presented contributions in bullet notes.

  5. Provide a flow chat for the proposed method.

Reply: we add 3.4 section to show the flow chat .

 6. As this work uses Deep learning for Power system studies, the following references may be useful for the literature study. https://doi.org/10.1002/er.5331, https://doi.org/10.1016/j.engappai.2020.104000, https://doi.org/10.17775/CSEEJPES.2020.02700

Reply: we read these literature and add reference to our paper.